# Combined Curcumin and Luteolin Synergistically Inhibit Colon Cancer Associated with Notch1 and TGF-β Signaling Pathways in Cultured Cells and Xenograft Mice

**DOI:** 10.3390/cancers14123001

**Published:** 2022-06-18

**Authors:** Rukayat Aromokeye, Hongwei Si

**Affiliations:** Department of Human Sciences, Tennessee State University, Nashville, TN 37209, USA; raromoke@tnstate.edu

**Keywords:** synergistic, colon cancer, luteolin, curcumin, combination, notch1, TGF-β-signaling pathways, kinases, xenograft mice

## Abstract

**Simple Summary:**

One of the significant issues of the anti-cancer effects of phytochemicals, bioactive compounds from foods, and other plants, is that the effective dosages of the phytochemicals are too high to be obtained by oral intake, particularly by food intake. The current study aimed to assess if the combination of two phytochemicals, luteolin (LUT) and curcumin (CUR), at low dosages where LUT or CUR alone has no significant effect, synergistically exerts anti-colon cancer. Our results show that combined LUT and CUR synergistically suppressed colon cancer in cultured cells and cell-derived xenograft mice, which may be associated with two possible molecular pathways. This study provides a practical approach to treating or preventing colon cancer in humans by consuming foods having high levels of luteolin and curcumin.

**Abstract:**

This study aimed to select a combination of curcumin and luteolin, two phytochemicals from food, at lower concentrations with a higher inhibitory effect on colon cancer growth and investigate possible molecular mechanisms of this anti-colon cancer effect. By pairwise combination screening, we identified that the combination of curcumin (CUR) at 15 μM and luteolin (LUT) at 30 μM (C15L30) synergistically suppressed the proliferation of human colon cancer CL-188 cells, but the individual chemicals had a little inhibitory effect at the selected concentrations. This result was also confirmed in other colon cancer DLD-1cells, suggesting that this synergistic inhibitory effect of C15L30 applies to different colon cancer cells. The combination C15L30 synergistically suppressed the wound closure (wound healing assay) in CL-188 cells. We also found that the combination of CUR and LUT (at 20 mg/kg/day and 10 mg/kg/day, respectively, IP injection, 5 days for 2 weeks) synergistically suppressed tumor growth in CL-188 cell-derived xenograft mice. Western blot results showed that protein levels of Notch1 and TGF-β were synergistically reduced by the combination, both in CL-188 cells and xenograft tumors. Tumor pathological analysis revealed that combined CUR and LUT synergistically increased necrosis, but the individual treatment with CUR and LUT had no significant effect on tumor necrosis. Therefore, combined curcumin and luteolin synergically inhibit colon cancer development by suppressing cell proliferation, necrosis, and migration associated with Notch1 and TGF-β pathways. This study provides evidence that colon cancer may be prevented/treated by consuming foods having high levels of luteolin and curcumin in humans.

## 1. Introduction

More than 290 different phytochemicals from fruits, vegetables, nuts, and olive oil are believed to be the primary reason why the Mediterranean diet can reduce the rates of various chronic diseases, including cancer [1,2]. However, the intake of each phytochemical from the Mediterranean diet is much lower than the amount used in cellular and animal studies [3]. One of the explanations is that multiple phytochemicals at very low levels from the Mediterranean diet exert synergistic effects. Indeed, there is a vast gap between the range of concentrations of phytochemicals typically used in cell culture models (at μM or higher) and the levels in the bloodstream (usually at nM) following consumption of typical doses in foods and supplements [4,5]. Moreover, it is impossible for humans to take the amount of the food to reach the required circulating levels of the phytochemical calculated from the results of studies using animals or cells (for instance, one person needs 25 kg of red wine to reach similar effects in mice) [6]. On the other hand, increasing studies show that combinations of a couple of phytochemicals synergistically improve osteoporosis [7], suppress obesity [8], and inhibit breast cancer [9]. Therefore, combining multiple phytochemicals may be a practical approach to combat cancer.

Curcumin (1E,6E)-1,7-bis(4-hydroxy-3-methoxyphenyl)-1,6-heptadiene-3,5-dione, CUR), is a bioactive compound from Curcuma longa L (turmeric), a common cooking dye and traditional medical plant. Luteolin (3’,4’,5,7-tetrahydroxyflavone, LUT) is a flavonoid in many commonly consumed vegetables, including thyme, Chinese celery, radicchio, and peppers [10]. Individual curcumin [11,12,13] and luteolin [14,15,16] are well-known anti-cancer reagents. However, the half-maximal inhibitory concentration (IC50) of individual LUT and CUR in cancer cells is 50 μM [17,18] and 4–50 μM [19,20], respectively. These high concentrations are not physiologically achievable where the highest plasma concentration of LUT, CUR, and its metabolites transiently reach 15 μM [21] and 0.05 μM [22], respectively. One of our approaches to narrow the concentration gap between in vitro studies and human studies is to combine two phytochemicals to synergistically inhibit cancer, while the individual phytochemicals do not have an anti-cancer effect at the selected dosages. Indeed, the combination of LUT and CUR synergistically inhibited breast cancer both in cultured cells and xenograft mice (separate manuscript).

In the present study, we screened phytochemicals to select combinations exerting a synergistic inhibitory effect on colon cancer using a cell proliferation assay. We identified a novel combination of LUT and CUR synergistically constrains colon cancer cell proliferation in CL-188 cells and DLD-1 cells and tumor growth in CL-188 cells-derived xenograft mice. At the same time, the individual LUT and CUR do not have this anti-cancer effect at the selective dosages both in vitro and in vivo. This synergistic anti-colon cancer effect by this combination involves regulating the Notch1 and transforming growth factor- beta (TGF-β) pathways as well as inducing necrosis both in cells and tumors. These results suggest that the combination of LUT and CUR is an efficient approach to treating colon cancer.

## 2. Materials and Methods

### 2.1. Cell Culture and Treatment Reagents

The colon cancer cell line CL-188 (LS174T) and DLD-1 used in this study were obtained from the American Type Culture Collection (ATCC). The CL-188 cells were maintained in Minimum Essential Media (MEM), and DLD-1 cells were maintained in Roswell Park Memorial Institute medium (RPMI 1640, 11875093, Thermo Fisher medium, Waltham, MA, USA). Media were supplemented with 10% fetal bovine serum (FBS), and 1% Penicillin (100 U/mL) and streptomycin (100 mg/mL) as an antibiotic source. The cells were incubated in an atmosphere of 5% CO_2_ at 37 °C. To evaluate the effect of combined curcumin and luteolin on cell proliferation of CL-188 and DLD-1 cell lines, the medium was switched to phenol-red-free medium, supplemented with 10% FBS and 1% penicillin/streptomycin. Phytochemicals (curcumin and luteolin) used in this study were purchased from Sigma Aldrich (St. Louis, MO, USA). Chemicals were dissolved in DMSO (Sigma, St. Louis, MO, USA. Cat#: 67-68-5) at 100 mM stock solution and diluted to the concentrations to be used in cell culture treatments.

### 2.2. Cell Proliferation Assay

CL-188 cells were seeded in a 12-well plate (6.4 × 10^4^ cells/well) in phenol-red-free medium essential medium with 10% FBS and 1% penicillin/streptomycin, and DLD-1 cells were in RPMI 1640 supplemented with 10% FBS and 1% penicillin/streptomycin. After overnight incubation, cells were treated with various concentrations of CUR and LUT or a combination of both (for instance, DMSO, LUT 30 μM, CUR 10 μM, CUR 10 μM + LUT 30 μM, CUR 15 μM, CUR 15 μM, and LUT 30 μM). The plate was incubated at 37 °C for 72 h. Images were photographed with the microscope; the cell proliferation assay was performed using the MTT vybrant assay kit according to the manufacturer’s instructions. In brief, the medium was aspirated from each well and replaced with 200 μL of fresh medium. Then, 50 μL of PBS-MTT was added to each well, after which the plates were incubated at 37 °C for 2 h. The SDS-HCl solution was added to each well, followed by incubation at 37 °C for 4 h. Absorbance was read at 570 nm using the synergy hybrid plate reader (BioTek Instruments Inc., Winooski, VT, USA. Part # 8041000). All experiments were separately repeated 5 times.

### 2.3. Pairwise Combinations Screening

Individual phytochemical was used for pairwise screening using a non-constant concentration of phytochemicals in a ratio 1:1 to achieve a pairwise combination matrix. Cells were seeded at 6.4 × 10^4^ cells per well in a 24-well plate, 64 wells were used each time with 64 different combinations of 8 concentrations of CUR (0, 1, 5, 8, 10, 12, 15, and 20 µM) and 8 concentrations of LUT (0, 1, 5, 10, 15, 20, 25, and 30 µM) to determine the optimum dose required to inhibit cell growth. DMSO concentration was normalized for each well. After 72 h of treatment, cell viability was measured using the MTT vibrant assay kit according to the manufacturer’s instructions. Experiments for the cell line were performed at least 4 times, and each combination has two replicates for each independent experiment. Growth rate inhibition (GRI) was determined, and the mean values of GRI for each combination treatment were displayed as a combination matrix plot. A combination index (CI) plot was used to evaluate the synergistic effect of combination pairs. CI = D1/(Dχ)1 + D2/(Dχ)2 > 1 indicates an antagonistic effect, CI = 1, indicates an additive effect, CI < 1 indicates synergistic effect. Where (Dχ)1 and (Dχ)2 represented concentrations of each drug alone to exert χ% effect, while (D)1 and (D)2 were concentrations of drugs in combination to elicit the same effect (CI plot or Chou–Talalay plot [23].The optimized doses of combination pair with high effect levels (Fa > 0.7) and low CI (<0.6) values were used for further studies.

### 2.4. Wound-Healing Assay

A wound-healing assay was used to assess changes in the migratory ability of cells as previously described [24]. The cells were seeded in 6-well plates in the medium containing 10% FBS and 1% P/S. Cells were allowed to grow until about 80% confluency as a monolayer. Linear wounds were made with a P20 pipette tip in each monolayer well. The cells were starved with serum-free medium overnight and incubated with 10 μg/mL mitomycin C for 2 h before the scratch assay, which inhibited mitosis of the cells and allowed us to distinguish migration from proliferation as reported [25]. The wells were then treated with various concentrations of curcumin and luteolin or a combination of both for 72 h (DMSO, LUT 30 μM, CUR 10 μM, CUR 10 μM + LUT 30 μM, CUR 15 μM, CUR 15 Μm + LUT 30 μM). The plates were returned to the incubator. Images of scratch were taken at 4× magnification after 0 h, 24 h, and 72 h. The extent of migration of each well was analyzed with ImageJ to calculate the area of the wound closed and expressed as a percentage of DMSO [24].

### 2.5. Xenograft Model Establishment and Treatment

Nine-week-old male nude BALB/c mice homozygous for Foxn (Cat#: 007850) were purchased from Jackson Laboratory (Bar Harbor, Maine). Mice were housed in an environmentally controlled (23 ± 2 °C; 12 h light/dark cycle) animal facility and provided free access to the Teklad global rodent chow diet (Harlen Indianapolis, Indiana).

CL-188 cells were maintained in MEM in T-75 flasks to reach 80% confluency. Cells were harvested in Hanks’ Balanced Salt Solution (HBSS, Corning cello, cat#: 20-023-CV) mixed with Matrigel (Corning, Cat#: 356234) (ratio 1:1). Then, 100 µL of cell mixture containing 2 × 10^6^ CL-188 cells was injected subcutaneously into the flank of the hind leg of each mouse. The tumors were measured using a digital caliper every 2 days throughout the experiment. Tumor volume was calculated using the formula π/6×L×W×H. Mice with similar body weight and tumor volume were assigned to the same treatment group. After 10 days of inoculation, mice with tumor volume averaging 200 mm^3^ in each group as described [26].

CUR and LUT (MilliporeSigma, Burlington, MA, USA) were dissolved in a vehicle containing 5% DMSO, 5% Tween20, and 90% PBS *v*/*v*%. Mice were assigned into one of four (4) groups to be treated with Vehicle, LUT 10 mg/kg body weight, CUR 20 mg/kg body weight, and a combination of LUT 10 mg/kg body weight + CUR 20 mg/kg body weight with a total of 9 mice per group. Then, 0.1 mL of chemicals were injected intraperitoneally in mice every 5 days, allowing 2 days of rest for 2 weeks. The experiment was terminated after 2 weeks, as mice in the vehicle group developed tumors with a volume of approximately 3000 mm^3^. Mice were euthanized as per the American Veterinary Clinical Affiliation (AVMA) guidelines, and tumors were excised, weighed, and stored for further analysis. Other tissues collected include the liver, serum, and colon. The maximum tumor size permitted by the Institutional Animal Care and Use Committee (IACUC) of 3000 mm^3^. Animal experiments were approved by the IACUC at Tennessee State University (Nashville, TN, USA), protocol No. 16-11-636, dated 6 March 2018.

### 2.6. Western Blot

For cells, harvested cells were lysed in mammalian protein extraction buffer (Cat #: 78501, Thermo-scientific, Waltham, MA, USA. 25 mM Tris-HCl, pH 7.6, 150 mM NaCl, 1% sodium deoxycholate, 0.1% SDS) and sonicated for 15 s (5 s thrice) on the ice at intervals. After centrifuging at 12.7 RPM for 5 min at 4 °C, the supernatant was removed, and protein concentration was measured using the Pierce BCA protein assay kit (Thermo Scientific, Waltham, MA, USA, Cat #: 23225). For tumors, the frozen mice tumors were cut, weighed (20–30 mg), and placed in 400 µL of mammalian protein extraction buffer. The tissue was cut into very tiny pieces and homogenized using a tissue homogenizer (Biospec Products Inc., Bartlesville, OK, USA. Model 985370) at a level of 20,000 RPM for 5 s thrice on ice at intervals. Tumor lysate was centrifuged at 12.7 RPM for 5 min at 4 °C (Thermo scientific., Waltham, MA, USA 75002446), and the medium part was used to measure protein concentration using the Pierce BCA protein assay kit. Based on the protein centration of the sample, each total protein was added with an appropriate sample buffer and heated at 95 °C for 5 min. Total protein concentration was loaded on the SDS-PAGE gel and transferred to a nitrocellulose membrane (GE Healthcare Life Sciences, Piscataway, NJ, USA). The membranes were blocked with 5% non-fat dry milk at room temperature for 1 h. Antibodies against NOTCH1 (D1E11), GAPDH (D16H11), and TGF-β (56E4) (purchased from Cell Signaling Technology, Danvers, MA, USA) were used for immunoblotting, all at a dilution factor of 1:1000. Membranes were incubated with secondary antibodies at room temperature for 1 h, and intensities of X-ray films were quantified using the ImageJ software. The NOTCH1 and TGF- β proteins in the experiment were normalized with GAPDH as an internal control to confirm that protein loading is equal across the gel as we described [9]. Please see original WB images in Appendix A.

### 2.7. Histological Analysis

Tumors excised from mice were fixed in 10% formalin. The tissues were processed by dehydrating, clearing, infiltration, embedding, and sectioning. These sections were then subjected to H&E staining [27]. Photomicrographs of representative regions of each tumor were captured using an Olympus BX41 microscope. An Olympus UC90 microscope-mounted camera and Olympus cellSens Standard 1.18 photomicrograph capturing software. Images were taken at 20× magnification and imported into the Java image processing software, ImageJ. Images were converted to 8bit in ImageJ and evaluated using the thresholding tool as described in the ImageJ user manual. Threshold levels were selected to measure all pixels in the selected image representing all tissue uptaking hematoxylin and eosin stain. A second threshold level was set to measure all pixels reaching an intensity threshold that represented a viable (non-necrotic) tumor. The difference in total pixels minus those whose threshold level matched that of the viable tumor was interpreted as necrosis, and a percentage was calculated.

### 2.8. Statistical Analysis

Experiments were repeated at least thrice for in vitro studies. Data were analyzed with one-way ANOVA and expressed as mean ± standard error. Paired T-tests were used to differentiate between mean where different. A significant difference was set at *p* < 0.05 (*), *p* < 0.01 (**), and *p* < 0.001 (***).

## 3. Results

### 3.1. Combination of Luteolin and Curcumin Synergistically Inhibited Human Colon Cancer Cell Proliferation in CL-188 Cells and DLD-1 Cells

Based on our recent screening of 20 phytochemicals, the combination of curcumin and luteolin synergistically inhibited breast cancer cell proliferation (separate manuscript). In the present study, we would like to investigate if the combination of curcumin and luteolin also synergistically inhibits colon cancer cell proliferation. Firstly, we conducted a time course (24 h, 48 h, and 72 h) to determine the optimum time with the best inhibitory effects of the combination in CL-188 cells and DLD-1 cells. We found that 72 h treatment has the most inhibitory effects of the combination in the two cell lines (Figure 1A). Next, eight concentrations of curcumin and luteolin were selected to generate a dose-response curve in CL-188 cells. The IC50 for luteolin and curcumin were 27.3 µM and 13.4 µM, respectively. Pairwise combination screening for the concentrations of luteolin (0, 1, 5, 10, 15, 20, 25, and 30 µM) and curcumin (0, 1, 5, 8, 10, 12, 15, and 20 µM) in 8 × 8 dose-response matrix was conducted to select the best combination. This combination should have a low combination index (CI) and has a high inhibitory effect, but the individual chemical had few inhibitory effects at the chosen concentrations. Figure 1B shows the heat map of the growth rate inhibition of combinations of CUR and LUT. The plots of combination index and inhibitory effects (Fractions of affected cells, Fa) are shown in Figure 1C. The combination (C15L30) of CUR at 15 µM and LUT at 30 µM with a low combination index of 0.25 and highest inhibitory effect/Fa value of (0.75) was selected as the optimum combination for further studies in CL-188 and DLD-1 cells. As shown in Figure 1D,E, the combination C15L30 has the best inhibitory effect (reduced cell numbers to 51% of DMSO control, *p* = 0.00012) in CL-188 cell proliferation, which is significantly lower than the number of the combination C10L30 (69.6% of DMSO, C10L30 vs. C15L30 *p* = 0.00043) and individual chemicals C15 (83.6% of DMSO, C15 vs. C15L30, *p* = 0.0011), C10 (91.4% of DMSO, C10 vs. C10L30, *p* = 0.0044), and L30 (92.0% of DMSO, L30 vs. C15L30, *p* = 0.0036). These results were also confirmed in other human colon cancer DLD-1 cells (Figure 1F,G) in which C15L30 synergistically inhibited cell proliferation to 31.4% of DMSO, *p* = 0.00014) compared to C10L30 (47.6% of DMSO, C10L30 vs. C15L30, *p* = 0.0056) and individual chemicals C15 (70.0% of DMSO, C15 vs. C15L30, *p* = 0.00308), C10 (77.3% of DMSO, C10 vs. C10L30, *p* = 0.00173, and L30 (84.1% of DMSO, L30 vs. C15L30, *p* = 0.0016). The reason for selecting these two cell lines to confirm the inhibitory effect of C15L30 is not selective in only one cell line, given that these two colon cancer cell lines have same (adenocarcinoma, having molecules MSI, BRAF, and PTEN) and different (morphology, original patient gender, molecules CIMP, PIK3CA, and TP53) features [28]. Therefore, the combination C15L30 really synergistically inhibited colon cancer cell proliferation both in CL-188 cells and DLD-1 cells.

### 3.2. Combination of Curcumin and Luteolin Prevented the Closure of the Induced Wound in CL-188 Cells

A wound-healing assay is a typical experiment to test the invasion and migratory potentials of cancer cells. CL-188 cells were seeded in a 6-well plate and allowed to confluence, then starved with serum-free medium overnight and incubated with 10 μg/mL mitomycin C for 2 h to inhibit mitosis of the cells, which can distinguish migration from proliferation. Wounds were made to the plate using a P20 pipette tip, followed by treatments. Images were taken at 0, 24, and 72 h to measure the area of the wound. As shown in Figure 2, the C10L30, C15, and C15L30 treatments significantly (*p* < 0.05) reduced the area closure compared to DMSO at 24 h. At 72 h, C15L30 synergistically reduced the wound area closure (1081% of DMSO, C15L30 vs. DMSO, *p* = 0.0001) in CL-188 cells, which is significantly higher than the number of the combination C10L30 (821% of DMSO, C10L30 vs. C15L30, *p* = 0.04) and individual chemicals C15 (636% of DMSO, C15 vs. C15L30, *p* = 0.001), C10 (243% of DMSO, C10 vs. C10L30, *p* = 0.0029), and L30 (208% of DMSO, L30 vs. C15L30, *p* = 0.002). These results indicate that the combination C15L30 synergistically inhibits cancer cell invasion and migration, the critical steps of cancer development.

### 3.3. Combination of Luteolin and Curcumin Suppressed Protein Expression of Notch1 and TGF-Beta in CL188 Cells

Given that Notch1 and TGF-β signalings play crucial roles in cancer cell migration, angiogenesis, and metastasis, we hypothesized that a combination of CUR and LUT treatment-induced growth inhibition through the regulation of Notch1 and TGF-β proteins in CL188 cells. To test this hypothesis, we measured the level of these proteins in treated CL-188 cells. As shown in Figure 3A, after 72 h of treatment, the combinations C10L30 and C15L30 synergistically reduced the level of Notch1 protein expression. C15L30 reduced the Notch1 protein level to 44% of DMSO (C15L30 vs. DMSO, *p* = 0.00073) in CL-188 cells, which is significantly lower than the number of the combination C10L30 (52% of DMSO, C10L30 vs. C15L30, *p* = 0.011) and individual chemicals C15 (68% of DMSO, C15 vs. C15L30, *p* = 0.0015), C10 (70% of DMSO, C10 vs. C10L30, *p* = 0.0032) and L30 (79% of DMSO, L30 vs. C15L30, *p* = 0.004). Similarly, C15L30 treatment synergistically reduced the TGF-β protein level in CL-188 cells (Figure 3B). These results align with the synergistic inhibitory effects of C15L30 in CL-188 cells (Figure 1), implying that the inhibitory effect of the combination C15L30 may be mediated by intervening with the Notch1 and TGF-β-signaling pathways.

### 3.4. Combination of Curcumin and Luteolin Synergistically Suppressed CL-188 Tumor Growth in Xenograft Mice and Attenuated Protein Expression of Notch1 and TGF-Beta in Tumors

To confirm our in vitro findings in animal studies, immunodeficient BALB/C nude mice were injected with 2 million CL-188 cells subcutaneously to generate a xenograft mice model. Treatments via intraperitoneal injection were given on day 12 and continued over 14 days. We observed that individual chemicals (CUR and LUT) did not exert any significant difference (*p* > 0.05) by decreasing the tumor volume when compared with the vehicle group. At the same time, the combination treatment was significantly different from the vehicle starting at day 5 (*p* < 0.05), significantly different from CUR (at day 5, 10, and 14, *p* < 0.05), and also significantly different from LUT (at day 5, and 10, *p* < 0.05), as shown in Figure 4A. After mice euthanization, tumors were excised and separated carefully with other tissues. As shown in Figure 4B, the average size of tumors from the combination CURLUT was significantly smaller than those in other groups with VEH 2030.3 mm^3^ (*p* = 0.031), LUT 1314.7 mm^3^ (*p* = 0.04), and CUR 1310.7 mm^3^ (*p* = 0.046) (Figure 4C). Similarly, compared with VEH, the average tumor weight was reduced by 37.8% in LUT (LUT vs. CURLUT, *p* = 0.045), 42.4% in CUR (CUR vs. CURLUT, *p* = 0.05), and 65.6% in CURLUT (VEH vs. CURLUT, *p* = 0.034) in Figure 4D, particularly, CURLUT was significantly different with LUT or CUR, indicating that the combination CURLUT has a synergistic effect in inhibiting tumor growth.

To investigate possible molecular mechanisms, we also measured the protein expression of Notch1 and TGF-β in tumors. As shown in Figure 4E, the combination of CUR and LUT also significantly reduced the expression of Notch1 protein to 52% of VEH (*p* = 0.00075), which is significantly lower than the individual chemicals, CUR (66% of VEH, *p* = 0.032) and LUT (80.5% of VEH, *p* = 0.013). The level of TGF-β protein was also significantly reduced in mice tumors (58% of VEH *p* = 0.001, which is significantly lower than the individual chemicals, CUR (79% of VEH, *p* = 0.02) and LUT (91% of VEH, *p* = 0.04), as in Figure 4F. Notably, these inhibitory effects by combination CURLUT in tumors are similar to the results in our in vitro results (Figure 3). These results suggest that the combination of curcumin and luteolin suppresses colon cancer development via regulation of Notch1 and TGF-β pathways.

### 3.5. Combination of Curcumin and Luteolin Synergistically Induced Necrosis in Xenograft Colon Tumors

To further investigate the mechanism of the anti-colon cancer effect by combined CUR and LUT, the tumor tissues from mice were subjected to hematoxylin and eosin stain after fixing with 10% formalin. Photomicrographs of the representative regions of each tumor slide were captured at 20× magnification and analyzed using Image J. As shown in Figure 5A, combined CUR and LUT promoted necrosis in tumor tissues (areas in asterisks) compared to the individual CUR or LUT. As shown in Figure 5B, the average necrosis of tumor tissues from the combination CURLUT was significantly higher than those in the LUT group (*p* = 0.0012), VEH group (*p* = 0.051), and CUR group (*p* = 0.07), indicating that the anti-colon cancer effect of combination CURLUT is at least partly by promoting necrosis in tumors.

## 4. Discussion

This study identified a novel combination of luteolin and curcumin that exerts synergistic anti-colon cancer effects both in vitro and in vivo. However, the individual luteolin and curcumin have much less or no such effects at the selected concentrations. This synergistic anti-colon cancer effect of the combination of luteolin and curcumin was observed in two human colon CL-188 cells and DLD-1 cells as well as in CL-188 cells-derived xenograft mice. Moreover, this synergistic effect of the combination of luteolin and curcumin was also observed in the protein expressions of Notch1 and TGF-β in CL-188 cells and xenograft tumors, which is in line with the anti-cancer effect in cells and tumors. We also found that this combination of luteolin and curcumin significantly induced necrosis in tumors. Therefore, the combination of luteolin and curcumin synergistically exerts anti-colon cancer through regulating Notch1 and TGF-β pathways and inducing necrosis.

While phytochemicals have been extensively investigated in colon cancer prevention and treatment, there is a big gap between effective concentrations of phytochemicals in cells (μM) and the physiological circulating levels of the phytochemicals in humans (nM), mainly via dietary supplementation of pure phytochemicals or foods [29]. Our solution for this significant gap is to combine two phytochemicals at relatively low concentrations to prevent/treat colon cancer. Therefore, the primary goal of this study is to identify a combination of two phytochemicals synergistically inhibiting colon cancer while the individual phytochemical does not have the inhibitory effect at the selected concentration. As aforementioned, the combination C15L30 is identified and worked in both in vitro studies and xenograft mice. In addition, we also found that the combination of luteolin and curcumin synergistically inhibited breast cancer both in cultured cells and xenograft mice (separate paper), indicating that the combination of luteolin and curcumin may synergistically inhibit all cancers; although, more studies are needed. Indeed, there are increasing reports that combined phytochemicals exert synergistic anti-cancer effects. For instance, Emulsome nanoformulations of curcumin (25 μM) and piperine (7 μM) effectively suppressed cell proliferation to about 50% viability in colorectal cancer HCT116 cells [30]. Combined curcumin and resveratrol inhibited proliferation in p53 positive and negative colon cancer HCT-116 cells [31].

Based on the increasing studies, we recently summarized five mechanisms to understand how a combination of two or more phytochemicals exerts synergistic effects in cells, animals, and humans [32]: (1) enhance the bioavailability of phytochemicals; (2) increase antioxidant capacity; (3) interact with gut microbiome (change microbial profiles, reduce endotoxin and increase gut integrity); (4) target same and/or different signaling pathways; and (5) apply two or more of these four mechanisms simultaneously. For instance, the combination C15l30 synergistically affected protein levels of Notch1 and TGF-β and the rate of necrosis in tumors, which are in line with its inhibitory effects in cells and tumors, indicating that the combination C15L30 may inhibit colon cancer by regulating Notch1 and TGF-β pathways. In addition, the combination C15L30 not only can inhibit colon cancer cell proliferation but also synergistically suppress the cancer migration and invasion (by wound healing assay) as well as necrosis (tumor tissue analysis). Therefore, the combination C15L30 may prevent/treat colon cancer through multiple approaches and mechanisms.

Our other study found that the combination of LUT and CUR synergistically inhibited breast cancer both in cultured cells and xenograft mice, which is in line with the results of the current study, indicating that the synergistic inhibition of the combination of LUT and CUR may be universal for several different cancers. Particularly, RNA-seq transcriptome analysis in breast cancer tumors found that Notch1 and TGF-β pathways are the top two pathways contributing to the synergistic inhibition by the combination of LUT and CUR, and the protein analysis of Notch1 and TGF-β in tumors matched the RNA changes (separate manuscript). Therefore, we measured the tumor protein levels of Notch1 and TGF-β in this study when we found that the combination of LUT and CUR synergistically inhibited colon cancer both in cultured cells and xenograft mice.

Several studies have linked the increase in proliferation of colon cancer cells to Notch1 signaling [33,34]; particularly, Notch1 regulates cell proliferative abilities and regulates apoptosis in cells [35]. Jagged-1, a ligand of Notch1, contributes to metastasis in colon cancer [36]. Therefore, downregulation of the Notch1 signaling pathway may be an excellent approach to colon cancer therapy. As documented by Wang et al., curcumin has Notch inhibiting activity that may be used to treat cancer stem cells and solid tumors [37]. Notch1 induced cyclin D1 and CDK2 activity, two key molecules of cell proliferation in cervical cancer cells [38]. In the current study, we demonstrated that the combination of curcumin and luteolin down-regulates the protein expression in cultured cells and xenograft tumors. Therefore, the inhibitory effect in colon cancer by combined curcumin and luteolin is at least due to Notch1 downregulation.

The disruption of TGF-β could lead to various diseases, including cancer. TGF-β signaling is highly overexpressed, leading to tumor angiogenesis, invasion, migration, and metastasis in cancers, including breast cancer [39]. Gold et al. reported curcumin and emodin down-regulated TGF-β signaling in cervical cancer cells [40]. Evidence suggests that curcumin-induced apoptosis and reversed EMT through the downregulation of the TGF-β-signaling cascade in pancreatic cancer [41] and breast cancer [42]. In our wound healing assay, combined curcumin and luteolin synergistically inhibit migration and invasion in vitro. Immunoblotting of TGF-β in cells and mice tumors showed that the TGF-β protein expression was downregulated by the combination of curcumin and luteolin. These results suggest that TGF-β may be responsible for inhibited invasion and angiogenesis in colon cancer by the combination of curcumin and luteolin in both in vitro and in vivo studies.

There are several limitations of this study: (1) if knocking out genes of Notch1 and TGF-β abolish the anti-colon cancer effects of the combination of luteolin and curcumin; (2) IP injection of chemicals is not a typical approach in nutrition research; (3) a comprehensive study of the molecular mechanisms of the anti-colon cancer effects of the combination of luteolin and curcumin is required. Therefore, the future studies will be: (1) to knock out Notch1 and TGF-β genes in cells to test if this knockout can abolish the anti-colon cancer effects of the combination of luteolin and curcumin; (2) chemicals will be dietarily supplemented to xenograft or chemically induced colon cancer mice to test the synergistic anti-colon cancer effects by the combination of luteolin and curcumin; (3) whole-genome RNA-sequencing and metabolomic analysis of tumors will figure out comprehensive mechanisms on how combined luteolin and curcumin synergistically inhibit colon cancer.

## 5. Conclusions

In conclusion, a combination of luteolin (30 µM) and curcumin (15 µM) was selected as the optimum combination for the study due to a low combination index of 0.25 and its highest synergistic inhibitory effect on the growth of two human colon cancer cell lines CL-188 and DLD-1. Consistent with in vitro results, intraperitoneal injection of luteolin at 10 mg/kg body weight and curcumin at 20 mg/kg body weight in BALB/C Foxn nude mice for a 2-week period synergistically inhibited CL-188 cell-derived tumor growth. Further analysis showed that the synergistic anti-colon cancer effect exhibited by curcumin and luteolin was mediated through the downregulation of the effector protein Notch1, to induce growth cycle arrest, promote apoptosis, and TGF-β signaling inhibits angiogenesis and invasion in vitro and in vivo. These data demonstrate that a combination of curcumin and luteolin exerts an anti-colon cancer effect through the modulation of Notch1 and TGF-β signaling pathways, well-known in cancer growth, invasion, and metastasis. Therefore, these results provide solid evidence that the consumption of specific foods rich in curcumin and luteolin may be a practical approach to prevent or treat colon cancer in humans after more studies.

## Figures and Tables

**Figure 1 cancers-14-03001-f001:**
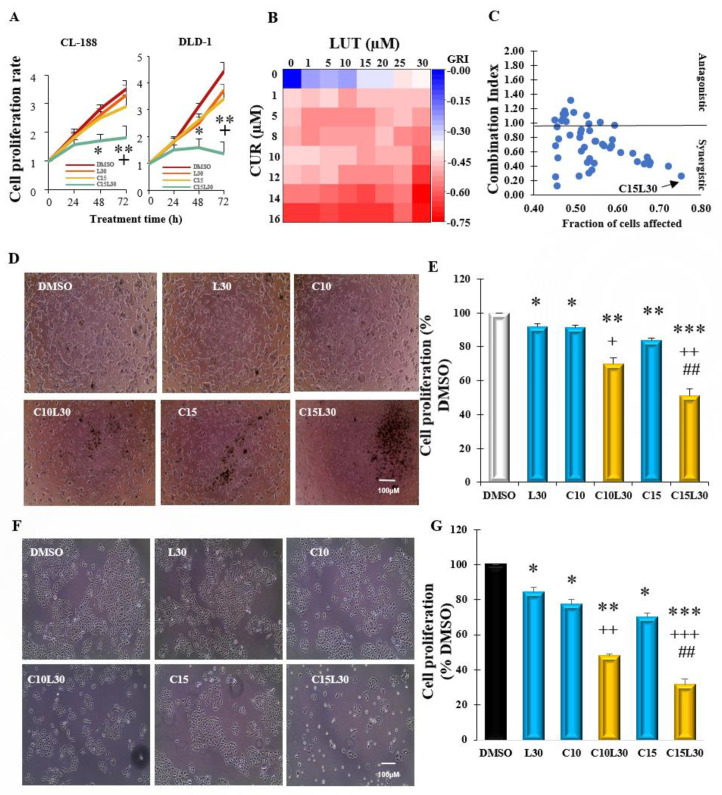
Combined curcumin and luteolin synergistically inhibited cell proliferation in colon cancer CL-188 cells and DLD-1cells: (**A**). The time course (24 h, 48 h, and 72 h) of cell proliferation by individual or combination of luteolin (LUT) and curcumin (CUR) in CL-188 and DLD-1 cells. (**B**). Heat map of combined CUR and LUT indicating inhibitory effects of the agents in CL-188 cells. (**C**). Quantitative plot of fraction affected-combination index showing the combinatorial doses of CUR and LUT in CL-188 cells. Each point represents the growth inhibition rate and fraction of affected cells. Points with a combination index (CI) > 1 indicates an antagonistic effect, CI = 1 indicates an additive effect, and CI < 1, a synergistic effect. (**D**). Representative images of CL-188 cells treated with LUT 30 µM, CUR at 10 or 15 µM, and their combinations for 72 h. (**E**). Bar graph showing growth inhibition of individual chemical CUR at 10 or 15 µM, LUT at 30 µM, or combinations in CL-188 cells. (**F**). Representative images of DLD-1 cells treated with LUT 30 µM, CUR at 10 or 15 µM, and their combinations for 72 h. (**G**). Bar graph showing growth inhibition of individual chemical CUR at 10 or 15 µM, LUT at 30 µM, or combinations in DLD-1 cells. *, Significant difference between DMSO and group; +, Significant difference between C10L30 and C10, L30 or C15L30 and C15 or L30; #, Significant difference between C10L30 and C15L30. *, + *p* < 0.05, **, ++, or ## *p* < 0.01, ***, +++ *p* < 0.001. Results of A, E, and G were from the average of 3–4 separate repeats.

**Figure 2 cancers-14-03001-f002:**
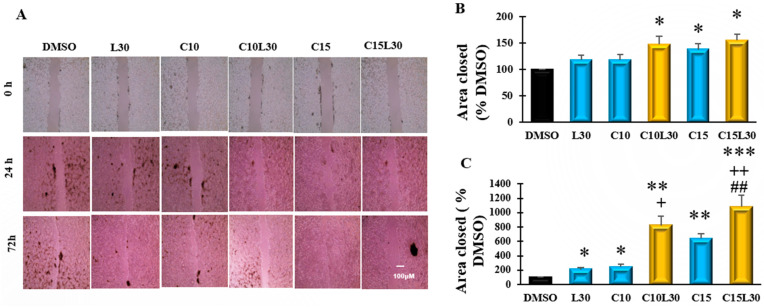
Combined curcumin (CUR) and luteolin (LUT) synergistically suppressed wound healing process in CL-188 cells: (**A**). Representative images of CL-188 cells treated with LUT 30 µM, CUR at 10 or 15 µM, and their combinations for 0 h, 24 h, and 72 h. (**B**). Bar graph showing area of wound closed in CL-188 cells using agents CUR at 10 or 15 µM, LUT at 30 µM, or combinations after 24 h. (**C**). Bar graph showing area of wound closed in CL-188 cells using agents CUR at 10 or 15 µM, LUT at 30 µM, or combinations after 72 h. Data were expressed in mean ± SEM of at least four independent experiments. *, Significant difference between DMSO and group, +, Significant difference between C10L30 and C10, L30 or C15L30 and C15 or L30; #, Significant difference between C10L30 and C15L30. *, + *p* < 0.05, **, ++, or ## *p* < 0.01, *** *p* < 0.001.

**Figure 3 cancers-14-03001-f003:**
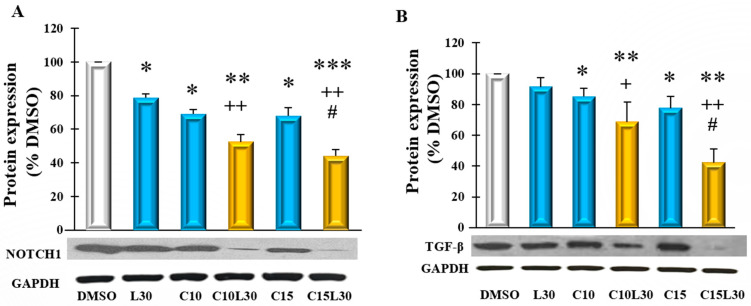
Combined curcumin (CUR) and luteolin (LUT) synergistically inhibited protein expression of Notch 1 and TGF-β in CL-188 cells: (**A**). Bar graph indicating the level of Notch1 in CL-188 cells treated with the CUR and LUT and their combinations after 72 h. (**B**). Bar graph indicating the level of TGF-β in CL-188 cells treated with the CUR and LUT and their combinations after 72 h. Data were expressed in means ± SEM of at least four independent experiments. *, Significant difference between DMSO and group, +, Significant difference between C10L30 and C10, L30 or C15L30 and C15 or L30; #, Significant difference between C10L30 and C15L30. *, +, # *p* < 0.05, **, or ++, *p* < 0.01, ***, *p* < 0.001. Please see original WB images in Appendix A.

**Figure 4 cancers-14-03001-f004:**
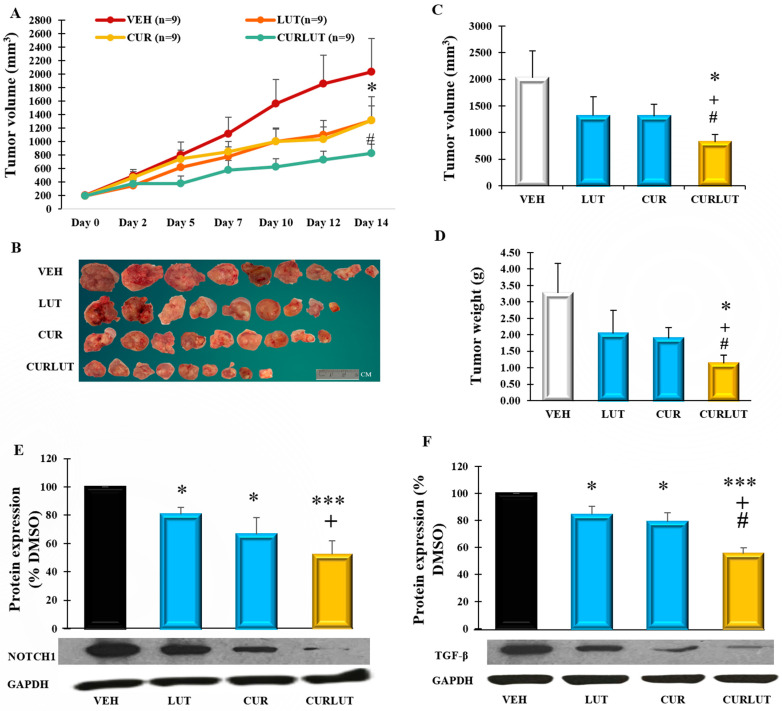
Combination of curcumin (CUR) and luteolin (LUT) suppressed colon cancer tumor growth in CL-188 cell-derived xenograft mice; 100 µL of CL-188 cells mixed with Matrigel and HBSS in ratio 1:1 was injected into the flank of BALB/C nude mice. Tumor volume was monitored thrice a week. Mice with similar body weight and tumor volume were assigned into one of four groups (average tumor volume of 200 mm^3^/group) to receive vehicle (VEH, 5% DMSO, 5% Tween20, 90% PBS), LUT (10 mg/kg/day), CUR (20 mg/kg/day) or the combination (CURLUT, LUT at 10 mg/kg/day + CUR at 20 mg/kg/day) via intraperitoneal injection daily, allowing 2 days rest for 2 weeks (14 days). (**A**). A line showing tumor volume with time across all treatment groups of VEH, LUT, CUR, and CURLUT. (**B**). Illustration of tumor sizes from all groups. (**C**). A bar graph showing the average tumor volume in mm^3^ across all treatment groups. (**D**). A bar graph showing the average tumor weight of mice expressed in grams (g) between groups. (**E**). Bar graph indicating the level of Notch1 in tumors of mice treated. (**F**). Bar graph indicating the level of TGF-β in tumors. Data were expressed as means ± SEM of the animals, *n* = 9 mice/group. * Significant difference between group and VEH group, +, significant difference between group and LUT group, #, significant difference between group and CUR group. *, +, # *p* < 0.05, ***, *p* < 0.001. Please see original WB images in Appendix A.

**Figure 5 cancers-14-03001-f005:**
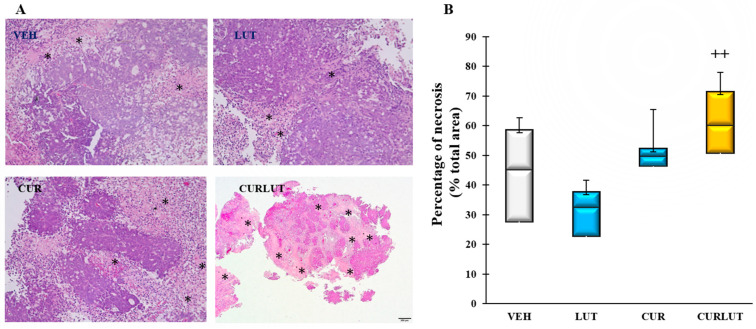
Combination of curcumin (CUR) and luteolin (LUT) synergistically induced necrosis in xenograft colon tumors: (**A**). Representative images of all groups. Areas of necrosis are pink-orange (asterisks); Intact tumor areas are dark pink-purple. 20× magnification. (**B**). A box plot showing necrosis of tumors (percentage of total area) within groups. + Significant difference between group and LUT group. ++ *p* < 0.01. *n* = 7–9.

## Data Availability

The study did not report any data.

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
