# Peer review of "Combined Curcumin and Luteolin Synergistically Inhibit Colon Cancer Associated with Notch1 and TGF-β Signaling Pathways in Cultured Cells and Xenograft Mice"

_cancers, 2022, doi:10.3390/cancers14123001_

Round 1

Reviewer 1 Report

The manuscript titled: “Combined curcumin and luteolin synergistically inhibit colon cancer via regulating Notch1 and TGF-β signaling pathways in cells and xenograft mice” is well written. However, the manuscript lacks the molecular mechanism to support the in vitro/ in vivo data observations. Data needs to be refined in a presentable manner. The present manuscript would be benefited by addressing the points below.

Major comments:

  1. In fig.1, authors have shown cell proliferation with MTT assay at 72 hrs; I request authors do MTT assay for cell growth at multiple time points (0, 24, 48, and 72 hrs.). The font in the figure is not clear, and please re-prepare the bar graphs with GraphPad Prism software.
  2. In fig.2, the authors have conducted a wound-healing assay without proliferation inhibitor. Confirming the wound closure is due to cell migration, not cell proliferation. The authors need to re-do the wound healing assay with cell proliferation inhibitors.    
  3. In fig. 3, authors have shown curcumin and luteolin down-regulate the expression of NOTCH1 and TGF-b in CL-188 cells. However, lack of data on the molecular mechanism related to their expression. Authors need to generate more data in this aspect.
  4. In fig. 5, the authors have shown that curcumin and luteolin suppress tumor growth by inducing necrosis in the tumor cells. However, in vitro data from the proliferation assay showed that curcumin and luteolin inhibit the proliferation of CL-188 and DLD-1 cells. If curcumin and luteolin induce the necrosis, I will request the authors to study the LDH cytotoxicity assay in vitro using CL-188 and DLD-1 cells. I request authors provide the IHC of ki-67, TGF-b, NOTCH1, and necrosis related markers data on tumor sections to confirm the in vitro observations.

Author Response

Q 1. In fig.1, authors have shown cell proliferation with MTT assay at 72 hrs; I request authors do MTT assay for cell growth at multiple time points (0, 24, 48, and 72 hrs.). The font in the figure is not clear, and please re-prepare the bar graphs with GraphPad Prism software.

Experiments of cell proliferation at 0, 24, 48, and 72 hrs were conducted, and results have been added as Fig.1A. Relevant results and descriptions were also added in lines 215-219 and 250-251 . All bar graphs and appropriate font and size were revised in figures 1-5.

Q 2. In fig.2, the authors have conducted a wound-healing assay without proliferation inhibitor. Confirming the wound closure is due to cell migration, not cell proliferation. The authors need to re-do the wound healing assay with cell proliferation inhibitors.   

The assay was re-done that cells were starved with serum-free medium overnight and incubated 10 μg/ml mitomycin C for 2 hrs prior to the scratch assay, which inhibited mitosis of the cells and allowed us to distinguish migration from proliferation as reported. The results were very similar to the previous results. This information has been added in Methods and Results, on lines 128-131 and 269-271.

Q 3.In fig. 3, authors have shown curcumin and luteolin down-regulate the expression of NOTCH1 and TGF-b in CL-188 cells. However, lack of data on the molecular mechanism related to their expression. Authors need to generate more data in this aspect.

We added a paragraph in the discussion to address this issue in lines 426-435 as “Our other study found that the combination of LUT and CUR synergistically inhibited breast cancer both in cultured cells and xenograft mice, which is in line with the results of the current study, indicating that the synergistic inhibition of combination of LUT and CUR may be universal for several different cancers. Particularly, RNA-seq transcriptome analysis in breast cancer tumors found that Notch1 and TGF-β pathways are the top two pathways contributing to the synergistic inhibition by the combination of LUT and CUR, and the protein analysis of Notch1 and TGF-β in tumors matched the RNA changes (separate manuscript). Therefore, we measured the tumor protein levels of Notch1 and TGF-β in this study when we found that the combination of LUT and CUR synergistically inhibited colon cancer both in cultured cells and xenograft mice.”

Q 4. In fig. 5, the authors have shown that curcumin and luteolin suppress tumor growth by inducing necrosis in the tumor cells. However, in vitro data from the proliferation assay showed that curcumin and luteolin inhibit the proliferation of CL-188 and DLD-1 cells. If curcumin and luteolin induce the necrosis, I will request the authors to study the LDH cytotoxicity assay in vitro using CL-188 and DLD-1 cells. I request authors provide the IHC of ki-67, TGF-b, NOTCH1, and necrosis related markers data on tumor sections to confirm the in vitro observations.

As we summarized in lines 421-424, combination C15L30 may prevent/treat colon cancer through multiple approaches and mechanisms from inhibiting proliferation, migration, and invasion as well as inducing necrosis. This phenomenon is not only observed in colon cancer, but also in breast cancer by the combination of luteolin and curcumin. We are doing a comprehensive study now to understand these multiple approaches and mechanisms and planned to a separate manuscript.

Reviewer 2 Report

Aromokeye and Si present an interesting manuscript in which they show that combination of curcumin and luteolin has a tumor suppressive effect both in wound healing assays, cell viability assay and more importantly in tumor xenografts. They extend their finding to show that reduced expression of Notch-1 and TGF-beta correlates with combination treatment of curcumin and luteolin, and conclude that reduce expression of both Notch-1 and TGF-beta is the mechanistic basis for their observations.

Comments:

  1. Without any direct evidence, the author’s conclusions that downregulation of the Notch-1 and TGF-beta receptors is the mechanistic basis in premature. Rationale for choosing these two pathways is weak. What other pathways were examined or considered? Was a more global search for gene expression alterations in the presence of curcumin and luteolin conducted? Direct experiments with Notch and/or TGF-beta inhibitors would bolster the author’s conclusions.

  1. Figure 1 and associate results. The authors use the terms cell viability, cell proliferation and growth inhibition interchangeably. These terms have distinct meaning. The authors should re-word this section carefully to accurately represent their experiments and results.

  1. Discussion of the properties of and differences between the CL-188 and DLD-1 cell lines used in the experiments should be included in either the methods or results.

Author Response

Q 1. Without any direct evidence, the author’s conclusions that downregulation of the Notch-1 and TGF-beta receptors is the mechanistic basis in premature. Rationale for choosing these two pathways is weak. What other pathways were examined or considered? Was a more global search for gene expression alterations in the presence of curcumin and luteolin conducted? Direct experiments with Notch and/or TGF-beta inhibitors would bolster the author’s conclusions.

We picked up these two pathways based on our other study that the combination of LUT and CUR synergistically inhibited breast cancer both in cultured cells and xenograft mice, which is in line with the results of the current study, indicating that the synergistic inhibition of the combination of LUT and CUR may be universal for several different cancers. Particularly, RNA-seq transcriptome analysis in breast cancer tumors found that Notch1 and TGF-β pathways are the top two pathways contributing to the synergistic inhibition by the combination of LUT and CUR, and the protein analysis of Notch1 and TGF-β in tumors matched the RNA changes (separate manuscript). This has been added in lines 426-435.

Q 2. Figure 1 and associate results. The authors use the terms cell viability, cell proliferation and growth inhibition interchangeably. These terms have distinct meaning. The authors should re-word this section carefully to accurately represent their experiments and results.

Thank you so much. The relevant areas have been corrected to proliferation.

Q 3. Discussion of the properties of and differences between the CL-188 and DLD-1 cell lines used in the experiments should be included in either the methods or results.

We added the same and different features of the two cell lines and cited references in Results, lines 241-244 as” The reason for selecting these two cell lines to confirm the inhibitory effect of C15L30 is not selective in only one cell line, given that these two colon cancer cell lines have same (adenocarcinoma, having molecules MSI, BRAF, and PTEN) and different (morphology, original patient gender, molecules CIMP, PIK3CA, and TP53) features [28].”

Round 2

Reviewer 1 Report

The authors have addressed all comments. The manuscript is improvised after revision, and the present version of the manuscript is in acceptable form.